# OpenReview forum: "AstaBench: Rigorous Benchmarking of AI Agents with a Scientific Research Suite"
_ICLR.cc/2026/Conference — ICLR 2026 Oral_

### Official Review · Reviewer_JXif · 2025-10-31

**Soundness:** 3
**Presentation:** 3
**Contribution:** 3
**Rating:** 6
**Confidence:** 4

**Summary:**

The paper introduces AstaBench, a benchmark suite and surrounding tooling intended to holistically evaluate agents for scientific research assistance. Contributions include: (i) a 2400+ problem task suite spanning literature understanding, coding/execution, data analysis, and end-to-end discovery; (ii) a standardized agent environment with date-restricted, production-grade literature search and a computational notebook to foster controlled comparison; (iii) a leaderboard and cost-accounting toolkit that normalizes API prices and reports efficiency; and (iv) a set of baseline agent classes (including several “Asta” agents) used to evaluate 57 agents across 22 classes. Key empirical finding: despite progress on some sub-skills, science assistance remains far from solved; even competitive agents struggle outside literature tasks, and end-to-end success rates remain low.

**Strengths:**

1. Ambitious scope & breadth. The suite spans four categories with explicit date cutoffs and tool restrictions, a step toward controlled, reproducible comparison in agent settings where tool access often dominates outcomes.

2. Standardized environment. The Asta Scientific Corpus (date-restricted search/snippet tools) plus the Computational Notebook provides an agent-friendly interface decoupled from bespoke agent code—useful for generalization across architectures.

3. Cost-aware leaderboard. Normalized, time-invariant cost accounting and reporting of confounders (openness, tooling) encourage more honest comparisons across agents/providers.

4. Comprehensive baselines. Evaluation across 57 agents / 22 classes (including ReAct, Smolagents, and specialized Asta agents) provides a broad empirical snapshot rather than narrow, self-selected comparisons.

5. Clear high-level findings. The paper documents that literature tasks are comparatively mature, while coding/execution, analysis, and end-to-end discovery remain weak—useful signal for the community.

**Weaknesses:**

1. Potential product-driven bias and external validity. Several tasks are “inspired by actual user requests to deployed Asta agents,” which risks distributional alignment with the authors’ own agent design and data pipelines. The paper could perhaps quantify how many tasks are product-derived, how they were sampled, and present held-out non-Asta sources to reduce bias?

2. Judging protocols and reliability. The work leans on a “Rubric & LLM Judge” but provides limited detail on prompting, calibration, adjudication, and human validation (e.g., double-blind spot-checks, disagreement resolution, inter-rater reliability). Perhaps more rigorous reporting (e.g., bootstrap CIs, per-task variance, adjudication rates) could be done to support “rigorous” claims?

3. Fairness of comparisons across tooling. Many agents are evaluated with custom or fully-custom tools, while others are restricted to the standard toolset. Even with “tooling” labels on the leaderboard, this remains an apples-to-oranges comparison. Stronger tool access ablations (standard-tools-only vs customized) could accompany headline numbers?

4. Domain skew & representativeness. The benchmark is weighted toward CS with relatively shallow non-CS coverage (e.g., limited biomed beyond LitQA2). For a “scientific research” suite, the domain diversity is modest. Perhaps the authors could provide a breakdown by field, difficulty, and reasoning type to justify claims of “holistic” science?

5. End-to-end metrics under-specified. The paper argues end-to-end discovery is hard (e.g., step-wise success compounding) but lacks task-level pass criteria, failure taxonomies, and human-rated scientific quality of the final reports/code for E2E-Bench(-Hard). Perhaps clarify grading and include expert review samples?

**Questions:**

My questions are reflected in the weaknesses I mentioned above.

---

> ### Author Response · Authors · 2025-11-20
>
> Thanks for the detailed and thoughtful review.
>
> (Note due to length limits, we have split this response into two parts)
>
> > Potential product-driven bias and external validity. Several tasks are “inspired by actual user requests to deployed Asta agents,” which risks distributional alignment with the authors’ own agent design and data pipelines. The paper could perhaps quantify how many tasks are product-derived, how they were sampled, and present held-out non-Asta sources to reduce bias?
>
>
> The ScholarQA and Paper-Finding tasks in particular are derived from user queries to online systems used to do the tasks (which are production-ready versions of the systems we use as the "Asta-*" baselines).  While it is possible that this gives some advantage to those agents, we note that:
> - This can be true regardless of whether a benchmark is product-derived; many papers simultaneously publish a new dataset together with a new method and are thus susceptible to some degree of co-optimization.  An advantage of deriving benchmarks based on user-facing products is that these benchmarks are at least on some level grounded in the queries that real people actually care about.
> - For literature search, we include LitQA2-FullText-Search (sourced from independent prior work) alongside the Paper-Finding task, and we can see in Table 12 that the Paper-Finder agent does well relative to other agents on both tasks, suggesting that the performance boost is likely not from co-optimization.
>
> > Judging protocols and reliability. The work leans on a “Rubric & LLM Judge” but provides limited detail on prompting, calibration, adjudication, and human validation (e.g., double-blind spot-checks, disagreement resolution, inter-rater reliability). Perhaps more rigorous reporting (e.g., bootstrap CIs, per-task variance, adjudication rates) could be done to support “rigorous” claims?
>
> Appendix E mentions the general procedure used for validating each task, and we have updated it with more detailed information about annotation/validation where applicable. Full details for tasks sourced from prior work can be found in their respective papers.  For PaperFindingBench the judgements for open-ended queries were spot-checked by authors on the validation set; Appendix E.2 includes details on how relevance criteria were selected for the judge. For E2E we added some details to Appendix F.9 where we analyze failures of agents and check judge scores (judged 92% correct on a validation set sample of 50 rubric items).  For ScholarQA-CS2, we've added more details in the Appendix E.3 under "Validation of ScholarQA-CS2"; we find moderate human–model agreement at the system level (Kendall τ = 0.467), which rises substantially to 0.800 when excluding Elicit outputs for which experts show systematic dispreference.
>
> We also include a measure of per-task variance (all single-task results are associated with ± 95% CIs based on the standard error across problems), and have updated the paper to also include CIs for the overall averaged scores in Table 11 (Appendix D).
>
> Finally, we will release our full code upon publication (which includes prompts and scorer implementations), as well as our online leaderboard which associates results with specific code commits and runtime logs (our evaluation framework tracks these automatically to ensure reproducibility for all submitted results).

---

> ### Author Response · Authors · 2025-11-20
>
> (continued from previous comment)
>
> > Fairness of comparisons across tooling. Many agents are evaluated with custom or fully-custom tools, while others are restricted to the standard toolset. Even with “tooling” labels on the leaderboard, this remains an apples-to-oranges comparison. Stronger tool access ablations (standard-tools-only vs customized) could accompany headline numbers?
>
> Almost all of the "fully custom" agents on the leaderboard are closed-source systems, so we unfortunately cannot do an apples-to-apples comparison where we modify their toolset.  This is one of the challenges we aim to highlight by reporting the tooling labels, as existing benchmarks often fail to highlight potential tooling differences.  By releasing AstaBench and clearly denoting the custom-tooled systems, we hope to encourage more closed-source providers to conduct experiments with standardized tools on our suite.
>
> For the "custom interface" systems: this label does not mean they used custom tools; rather, they used tools based on the same APIs or with a comparable sandbox environment (see Appendix B) but with a custom implementation. In theory they should behave the same way as the standard ones, but we denote it as a separate category because there is always a possibility for some implementation details to be missed when attempting to replicate the tools (request parameters/rate-limiting/error-handling and so on).
>
> > Domain skew & representativeness. The benchmark is weighted toward CS with relatively shallow non-CS coverage (e.g., limited biomed beyond LitQA2). For a “scientific research” suite, the domain diversity is modest. Perhaps the authors could provide a breakdown by field, difficulty, and reasoning type to justify claims of “holistic” science?
>
> It's true that CS is in the majority, but we want to emphasize that we do have many tasks that cover other domains.  Shown in Table 2, we have Biology tasks as well as several tasks from mixed domains which include meta-science, social sciences, and medicine, humanities, economics, and engineering (we have updated the Appendix E sections to include the full domains for Mixed tasks).
>
> > End-to-end metrics under-specified. The paper argues end-to-end discovery is hard (e.g., step-wise success compounding) but lacks task-level pass criteria, failure taxonomies, and human-rated scientific quality of the final reports/code for E2E-Bench(-Hard). Perhaps clarify grading and include expert review samples?
>
> This is a good suggestion, we have added details in F.9. The three-way report/code/artifact rubric grading was designed to improve grading robustness (e.g., where a report makes a claim not substantiated in code), the grading has been clarified and quantified in Appendix F.9 and Table 20, along with more systematic human validation of LLM-as-judge scores (F.9). Task-level success is defined as satisfying *all* the required rubric criteria, rarely met by the E2E systems. Scores (max was 5%) are now reported in E.9 Table 19.

---

### Official Review · Reviewer_LRec · 2025-10-31

**Soundness:** 4
**Presentation:** 3
**Contribution:** 4
**Rating:** 8
**Confidence:** 4

**Summary:**

The paper introduces AstaBench, a suite of benchmarks for evaluation AI agents' potential for conducting scientific tasks. The suite aggregates scientific tasks and the authors run a comprehensive evaluation of different models on these tasks with different agent scaffolds. The suite controls for several factors, such as which tools are used, and provides a set of agent baselines that the authors run these evaluations on.

The paper's contributions to AI for science evaluations are impressive. Evaluating agents is a challenging problem, and the authors make a serious and compelling effort in that direction. I recommend acceptance, and offer suggestions for further improving the paper below.

**Strengths:**

- The authors calculate the cost for all evaluations, which is key for reliable benchmarking. Most existing frameworks don't do this (as the authors correctly identify).
- The scope of benchmarks is broad, including agents from many stages of the scientific research pipeline
- The number of models and agent scaffolds tested is impressive
- The insights from the analysis are solid; I appreciate the focus on in-depth analysis and insights as well as the focus on openly releasing all of the materials from the analysis

**Weaknesses:**

- Some parts in comparisons with other frameworks are overclaimed. For example, as far as I know, frameworks like HAL and Inspect do have open agents and account for variation in tool use/agent scaffolds.
- Similarly, some of these frameworks have support for general agents too, for example the Inspect ReAct agent or the HAL generalist agent that are available across benchmarks
- Claims like benchmarks in other suites not being "product informed" also seem like a stretch, especially since the other frameworks also incorporate many of the same benchmarks
- CORE-Bench has 45 tasks in train and 45 in validation; why does the table say 35 and 37?
- Could you clarify how the budget or number of steps for agents is set, especially over different models? How do you avoid infinite loops?
- In figure 2, how are the 11 benchmarks collapsed into 5 figures? What is the aggregation function?

**Questions:**

Overall, I think the paper is a great contribution and would be a strong fit for ICLR. I am happy to recommend acceptance. Moreover, I would raise my score further and recommend a spotlight or oral ***if the following points are addressed***:
- Please make sure the text of the paper matches the contributions of the paper. The results in the paper speak for themselves; there is no need to claim this is the "first" suite to do many of these things (in particular with the comparisons against past agent frameworks)
- Please clarify how you selected the agents you picked for the analysis. What other agents did you consider? Why did you select this subset?
- Please update the text to clarify the contributions based on my concerns in the weaknesses section (differences in the number of tasks for benchmarks, aggregation of results across benchmarks, and more generally updating the clarity of the writing and presentation).

---

> ### Author Response · Authors · 2025-11-20
>
> Thanks for the detailed and thoughtful review.
>
> > Some parts in comparisons with other frameworks are overclaimed. [...] some of these frameworks have support for general agents too [...] Claims like benchmarks in other suites not being "product informed" also seem like a stretch
>
> > Please make sure the text of the paper matches the contributions of the paper. The results in the paper speak for themselves; there is no need to claim this is the "first" suite to do many of these things (in particular with the comparisons against past agent frameworks)
>
> This is a fair point; we got carried away, and will rein in the comparisons and claims.  With that said:
> - Frameworks like HAL and Inspect do have open agents, but they do not standardize e.g. date restrictions for literature search tools, and Inspect also doesn't report cost (while it does track the number of tokens which are used, it doesn't map those to dollar costs, which is extremely important when comparing across models).
> - While we don't claim to be the only product-informed benchmark, we still note that most benchmarks do not include tasks based on real product usage, whereas we contribute two newly-created ones (ScholarQA-CS2 and PaperFindingBench).
>
> > CORE-Bench has 45 tasks in train and 45 in validation; why does the table say 35 and 37?
>
> For AstaBench, we removed the problems that require a GPU to run, as otherwise the resource requirements for running the benchmark would be quite expensive (especially when running in parallel). This detail was previously buried in the Appendix (E.6), and given the potential for confusion, we now use a separate name, calling the AstaBench version CORE-Bench-Hard⁻ (superscript -) and making the modification more prominent.
>
> > Could you clarify how the budget or number of steps for agents is set, especially over different models? How do you avoid infinite loops?
>
> Following other studies that use ReACT, we heuristically set a fixed upper limit of steps to avoid infinite loops, which for all tasks is conservatively set to 100 steps. We note that this is more than enough to solve most tasks and that agents in practice use much less by submitting their answers long before this limit. We found that most models with ReACT and all models tested with Smolagents submit within an average of less than 12 steps. The exceptions are ReAct with Llama-scout (72 steps) and Gemini-flash (26 steps), which do sometimes devolve into looping behavior when they cannot figure out how to make progress.
>
> > Please clarify how you selected the agents you picked for the analysis. What other agents did you consider? Why did you select this subset?
>
> We focused on two simple, distinct and well-known architectures for our generalist baselines (ReAct, and the "CodeAct"-like Smolagents CodeAgent).  For task-specific agents, we aimed to include our own hand-designed systems as well as any prominent baselines known to do well (this included inviting many commercial tools to participate, although not all accepted).  There are of course many more baselines we'd love to include (Magentic-ONE, HAL agents, retry/escalation techniques, etc), as well as further optimization of existing baselines (DSPy prompt optimization or model finetuning), but ultimately we had to draw the line based on time and computation constraints.
>
> > In figure 2, how are the 11 benchmarks collapsed into 5 figures? What is the aggregation function?
>
> We aggregate according to the "task category" (the mapping is in Table 2, and Appendix D includes the full results for the individual tasks).

---

### Official Review · Reviewer_zo8M · 2025-11-01

**Soundness:** 4
**Presentation:** 4
**Contribution:** 4
**Rating:** 8
**Confidence:** 4

**Summary:**

Paper features an AstaBench - benchmark for evaluating scientific LLM agents on the variety of tasks. It integrates multiple benchmark for each task and consists not only with raw data, but with tools essential for testing. Benchmark can be used for testing agents on individual tasks (e.g. literature review or data analysis) as well as on end-to-end research.

**Strengths:**

- AstaBench provides a set of tools for testing agents, which makes it easier to setup experiments.
- Benchmark can be used for testing agents for specific tasks as well as for end-to-end research tests.
- AstaBench have a potential to become strong and stable benchmark for testing scientific LLM agents.

**Weaknesses:**

- Currently benchmark is heavily weighted for computer science and machine learning domains

**Questions:**

- Could you please provide at least brief description of scoring metrics in the main part of the paper? It's an important part of the benchmark, yet it's currently hidden in the appendix.
- It probably would be better to replace "AI" with "LLM" in title and abstract as benchmark made mainly for LLM agents, not for any AI agents (which also may refer to RL and other areas).

---

> ### Author Response · Authors · 2025-11-20
>
> Thanks for the detailed and thoughtful review.
>
> > Currently benchmark is heavily weighted for computer science and machine learning domains
>
> It's true that CS is in the majority, but we want to emphasize that we do have many tasks that cover other domains.  Shown in Table 2, we have Biology tasks as well as several tasks from mixed domains which include meta-science, social sciences, and medicine, humanities, economics, and engineering (we have updated the Appendix E sections to include the full domains for Mixed tasks).
>
> > Could you please provide at least brief description of scoring metrics in the main part of the paper? It's an important part of the benchmark, yet it's currently hidden in the appendix.
>
> We have done our best to include a brief description and to make the pointer to the appendix more prominent. If you have a suggestion of what we might cut, we could use that additional space to elaborate?
>
> > It probably would be better to replace "AI" with "LLM" in title and abstract as benchmark made mainly for LLM agents, not for any AI agents (which also may refer to RL and other areas).
>
> Can you explain your logic?  We created AstaBench to be a general;-purpose test of AI scientific reasoning, which is independent of a particular agentic technology. We believe it to be agnostic to any agent-implementation technology, whether LLM-based, search-based or trained via RL.

---

### Official Review · Reviewer_zxpV · 2025-11-04

**Soundness:** 3
**Presentation:** 3
**Contribution:** 3
**Rating:** 6
**Confidence:** 4

**Summary:**

This paper puts forward outstanding issues in the evaluation practices for AI science-specific agents. They introduce AstaBench, an evaluation suite that provides a holistic measure of agentic ability to perform scientific research.

**Strengths:**

- The paper does identify open issues in agent evaluations and provides a thorough analysis of the field. The paper provides a holistic evaluation of various models and scaffolds.
- The set of benchmarks included in the analysis is broad and offers good coverage of the AI science benchmark domain
- The writing is clear and overall easy to parse.

**Weaknesses:**

- Treating tools as confounding variables only makes sense when we want to collapse agent evals to model evals. For measuring how well agents with the latest model-backbones can do on a task, we should compile each model with the best-possible scaffold around it. This often involves using the tools the agentic models was e.g. trained with natively (web search, code interpreter, etc.)
- Other confounders aside from standard tools have been motivated and described in previous work [1]
- The Pareto efficiency frontier should be convex because every point on the convex envelope can be achieved by mixing between two agents.

[1] https://arxiv.org/abs/2407.01502

**Questions:**

Why are non-standard tools a confounding variable if we don't focus on comparing models but agent performance?

---

> ### Author Response · Authors · 2025-11-20
>
> Thanks for the detailed and thoughtful review.
>
> > Treating tools as confounding variables only makes sense when we want to collapse agent evals to model evals. For measuring how well agents with the latest model-backbones can do on a task, we should compile each model with the best-possible scaffold around it. This often involves using the tools the agentic models was e.g. trained with natively (web search, code interpreter, etc.)
>
> > Why are non-standard tools a confounding variable if we don't focus on comparing models but agent performance?
>
> We agree that different researchers may wish to study different aspects of AI agency, including 1) model performance (holding agent architecture and tools constant), 2)  agent architecture (eg relative performance of ReAct vs MagenticOne each powered by the same LLM and using the same tools), and 3) the best way to achieve high performance (possibly varying LM, agent architecture AND tools), amongst other objectives. AstaBench supports *all* of these goals. By providing and highlighting standard tools we facilitate 1 & 2 but still allow and support 3.  Of course, researchers who endow their agent with general Web search etc will have trouble with reproducibility and assigning credit to performance differences given that the state of scientific knowledge changes over time. In contrast, AstaBench’s literature search tools include date controls to enhance reproducibility.
>
> > Other confounders aside from standard tools have been motivated and described in previous work [1]
>
> Thanks for mentioning that work; it strongly influenced our thinking, but the citation was accidentally cut in the process of shortening the paper for ICLR.  We have added back this important reference!
>
> > The Pareto efficiency frontier should be convex because every point on the convex envelope can be achieved by mixing between two agents.
>
> This is a great point, thanks. We’ve updated the paper to use the convex hull when identifying pareto-optimal agents.

---

### Author Response · Authors · 2025-11-20

We'd like to thank all the reviewers for their many helpful comments.  We respond to specific questions individually (below), and we have also uploaded a revised PDF with the following high-level edits:

* Toned down our claims slightly.
* Switched to using convex hull as the Pareto frontier instead of stepwise best-at-cost.
* Added many additional details in Appendix E about the construction and verification of benchmarks, especially for ScholarQA-CS2 and E2E-discovery tasks.
  * This includes domain coverage information for the Mixed-domain tasks from Table 2
* Clarified the modifications to CORE-bench in the main text.
* Added CIs for the "Overall" scores
* Slightly adjusted paper-finder task instructions (and corresponding results) to more clearly represent the scoring criteria; we identified a risk that agents could be penalized for outputting too few results.  After the change, ReAct and Smolagents improve slightly on average but our main conclusions are unchanged.
* Some minor clarifications and rewording throughout; additional citations.

---

### Meta-Review · Area_Chair_mShh · 2026-01-07

**Summary:**

The paper presents AstaBench, a scientific research benchmark for evaluating AI agents rigorously and comprehensively. The benchmark contains more than 2400 problems spanning literature understanding, coding/execution, data analysis, and end-to-end discovery. To ensure evaluation fairness, the authors created two standardized tools: a date-restricted, production-ready literature search tool and a Jupyter notebook environment for coding. Using these environments, they compared 57 agents (both open and closed-source as well as general or custom to each task) in terms of both performance and cost. Through these empirical efforts, the authors show that current agents still struggle in many scientific tasks and end-to-end success is still very low.

The most important reviewer concerns are the following:

1- Overclaims

The authors admitted some level of exaggerated claims and reined them back in their new version, contextualizing their benchmark more carefully.

2- Is standardizing tool-use fair?

The standard tooling does not constraint the use of agents with other tools/scaffold, it just allows for more flexibility in testing both agents and models more rigurously.

3- Clarifications

The authors successfully addressed all of the clarifying questions and added content where needed.

**Reviewer Concerns:**

R1:
- Why are non-standard tools a confounding variable when focusing on agent (not model) performance?
	- The benchmark enables flexibility to focus on either agent or raw model comparisons by created a standard toolset.
- The parent efficiency frontier should be the convex hull since agent mixing is possible.
	- Proposal was accepted and added to paper.

R2:
- CS and ML dominate benchmark
	- This is true but many other domains such as biology, meta-science, social science, economics, medicine, humanities and engineering are also present
- Make score metric description more visible
	- Not enough space

R3:
- Overclaim: Other frameworks have general agents and account for tool use (HAL & Inspect)
	- They will tone down the comparison but HAL & Inspect don’t have date restrictions and Inspect doesn’t measure cost.
- Overclaim: Only benchmark to be product-informed is a stretch
	- It’s true that most benchmarks do not contain tasks from a real product unlike their ScholarQA-CS2 and PaperFindingBench, which come from real product-usage.
- Clarify CORE-Bench size
	- Some tasks were removed because they needed GPU’s, this will be clarified.
- Avoid infinite loops
	- 100 maximum steps based on previous works (most models never reach this threshold)
- Clarify aggregation of scores
	- Table 2
- Clarify agents choice criteria
	- 1) Simple, distinct and well-known general baselines like ReAct and CodeAct, 2) task-specific systems hand-designed by authors and 3) strong prominent baselines (many commercial tools were invited).

R4:
- Bias from product-derived tasks
	- Biasing one’s solution to be better at one’s dataset is something that happens regardless of whether it is product-informed or not.
- Clarify Rubric & LLM Judge process
	- Reviewer was pointed towards different Appendix sections and authors added more detailed explanations to existing sections.
- Fairness of comparison across tooling
	- This is precisely what the standard tools are attempting to achieve but it is useful to know how far custom or closed-source tools can take us.
- CS and ML dominate benchmark
	- This is true but many other domains such as biology, meta-science, social science, economics, medicine, humanities and engineering are also present
- More detailed breakdown of end-to-end metrics
	- Details were added to section F.9

In my opinion, all of the reviewer’s concerns were adequately addressed.

**Reviewer Scores:**

- zxpV: 6 -> 6
	- Reviewer doesn’t seem excited about the work even if concerns were addressed.
- zo8M: 8 -> 8
- LRec: 8 -> 10
	- Spotlight due to mention of oral/spotlight and rebuttal quality,
- JXif: 6 -> 8
	- All concerns were addressed and reviewer seems excited by the paper.

---

### Decision · Program_Chairs · 2026-01-26

Accept (Oral)